# Erythropoietin, Fibroblast Growth Factor 23, and Death After Kidney Transplantation

**DOI:** 10.3390/jcm9061737

**Published:** 2020-06-04

**Authors:** Michele F. Eisenga, Maarten A. De Jong, David E. Leaf, Ilja M. Nolte, Martin H. De Borst, Stephan J. L. Bakker, Carlo A. J. M. Gaillard

**Affiliations:** 1Division of Nephrology, Department of Internal Medicine, University of Groningen, University Medical Center Groningen, 9700 RB Groningen, The Netherlands; m.a.de.jong03@umcg.nl (M.A.D.J.); m.h.de.borst@umcg.nl (M.H.D.B.); s.j.l.bakker@umcg.nl (S.J.L.B.); 2Division of Renal Medicine, Brigham and Women’s Hospital, Harvard Medical School, Boston, MA 02215, USA; deleaf@bwh.harvard.edu; 3Department of Epidemiology, University of Groningen, University Medical Center Groningen, 9700 RB Groningen, The Netherlands; i.m.nolte@umcg.nl; 4Department of Internal Medicine and Dermatology, University of Utrecht, University Medical Center Utrecht, 3508 GA Utrecht, The Netherlands; C.A.J.M.Gaillard@umcutrecht.nl

**Keywords:** erythropoietin, fibroblast growth factor 23, death, kidney transplantation

## Abstract

Elevated levels of erythropoietin (EPO) are associated with an increased risk of death in renal transplant recipients (RTRs), but the underlying mechanisms remain unclear. Emerging data suggest that EPO stimulates production of the phosphaturic hormone fibroblast growth factor 23 (FGF23), another strong risk factor for death in RTRs. We hypothesized that the hitherto unexplained association between EPO levels and adverse outcomes may be attributable to increased levels of FGF23. We included 579 RTRs (age 51 ± 12 years, 55% males) from the TransplantLines Insulin Resistance and Inflammation Cohort study (NCT03272854). During a follow-up of 7.0 years, 121 RTRs died, of which 62 were due to cardiovascular cause. In multivariable Cox regression analysis, EPO was independently associated with all-cause (HR, 1.66; 95% CI 1.16–2.36; *P* = 0.005) and cardiovascular death (HR, 1.87; 95% CI 1.14–3.06; *P* = 0.01). However, the associations were abrogated following adjustment for FGF23 (HR, 1.28; 95% CI 0.87–1.88; *P* = 0.20, and HR, 1.45; 95% CI 0.84–2.48; *P* = 0.18, respectively). In subsequent mediation analysis, FGF23 mediated 72% and 50% of the association between EPO and all-cause and cardiovascular death, respectively. Our results underline the strong relationship between EPO and FGF23 physiology, and provide a potential mechanism underlying the relationship between increased EPO levels and adverse outcomes in RTRs.

## 1. Introduction

Renal transplant recipients (RTRs) have a high residual risk of all-cause and cardiovascular death, compared to the general population [1]. Previous studies demonstrated an independent association between higher circulating endogenous erythropoietin (EPO) levels and risk of all-cause and cardiovascular death among RTRs, similar to other patient populations such as chronic heart failure patients and the elderly [2,3,4]. In addition, administration of exogenous EPO may increase the risk of cardiovascular events in patients with chronic kidney disease (CKD) and end stage renal disease (ESRD) [5,6]. However, the underlying mechanisms responsible for the link between endogenous and exogenous EPO and adverse outcomes are unknown.

Studies from our group and others suggest that EPO is prominently involved in fibroblast growth factor-23 (FGF23) physiology [7,8,9,10]. FGF23 is an osteocyte-derived hormone that plays an essential role in regulating phosphate and vitamin D metabolism. In RTRs, increased FGF23 levels post-transplant are independently associated with an increased risk of graft failure and death [11,12]. Hypoxia, the main stimulus for EPO synthesis, stabilizes hypoxia-inducible factor (HIF)-1α, which is a heterodimeric transcription factor that regulates oxygen homeostasis [13,14]. Subsequently, stabilized HIF1-α upregulates FGF23 production while concomitantly increasing FGF23 cleavage into inactive fragments, resulting in elevated total FGF23 levels but normal levels of intact, bioactive FGF23 [15,16,17,18]. 

In the current study, we hypothesized that the previously established, but hitherto unexplained association between EPO levels and adverse outcomes may be attributable to increased levels of total FGF23. Therefore, we investigated the associations between EPO and total FGF23 levels and prospective outcomes in our RTRs cohort. 

## 2. Methods

### 2.1. Patient Population

All RTRs (aged ≥ 18 years) who were at least 1-year post-transplantation were approached for participation in the current study during outpatient clinic visits between 2001 and 2003. All RTRs were transplanted in the University Medical Center Groningen (Groningen, the Netherlands). The study has been described in detail previously [19]. Among 847 RTRs approached for participation, 606 RTRs agreed to participate and were included. All patients provided written informed consent and the study protocol was approved by the local medical ethical committee (METc 2001/039). The study protocol adhered to principles of the Declaration of Helsinki and the Declaration of Istanbul. The co-primary endpoints of the study were all-cause and cardiovascular death. Cause of death was obtained by linking the number of the death certificate to the primary cause of death as coded by a physician from the Central Bureau of Statistics according to the International Classification of Diseases, 9th revision (ICD-9; https://icd.codes/icd9cm). CV death was defined as deaths in which the principal cause of death was cardiovascular in nature, using ICD-9 codes 410 to 447. Secondary endpoint constituted death-censored graft failure (DCGF). DCGF was defined as return to dialysis or re-transplantation. For the current analyses, we excluded RTRs who did not have plasma samples available for measuring EPO levels (*n* = 14) and RTRs who used exogenous EPO (*n* = 13) due to positive interference in EPO measurement, resulting in 579 RTRs eligible for analyses. Median follow-up time from inclusion to endpoint was 7.0 (interquartile range (IQR), 6.2 to 7.4) years. Data on the co-primary and secondary endpoints were available in all 579 participants. There was no loss-to-follow-up in the current study. 

### 2.2. Data Collection

Relevant donor, recipient, and transplant characteristics at baseline were extracted from the Groningen Renal Transplant Database, as described in detail previously [19]. Information on medical history and medication use was obtained from patient records. Participants’ height and weight were measured with participants wearing light indoor clothing without shoes. Blood pressure was measured according to a strict protocol as previously described [19]. Alcohol consumption and smoking behavior were recorded using a self-reported questionnaire. Smoking behavior was classified as never, former, or current smoker. 

### 2.3. Laboratory Procedures

Blood samples were drawn during the next outpatient clinic visit after agreeing to participate. Blood was drawn in the morning after an 8–12 h overnight fast, and all measurements were performed in samples of the same timepoint. In plasma EDTA samples frozen at −80°C, we measured plasma EPO levels using an immunoassay based on chemiluminescence (Immulite, Los Angeles, CA) [20]. We measured plasma total FGF23 levels with a human FGF23 (C-terminal) enzyme-linked immunosorbent assay (ELISA; Quidel Corp., San Diego, CA, USA) with intra-assay and interassay coefficients (CVs) of variation of <5% and <16% in blinded replicated samples, respectively [21]. The total FGF23 immunometric assay uses two antibodies directed against different epitopes within the C-terminal part of FGF23, and as such the assay detects both the intact hormone as well as C-terminal cleavage products, and therefore measures total FGF23 levels. We measured plasma ferritin levels using an electrochemiluminescence immunoassay (Modular analytics E170, Roche diagnostics, Mannheim, Germany). Renal function was determined by estimating GFR by applying the Chronic Kidney Disease Epidemiology Collaboration equation [22]. Proteinuria was defined as urinary protein excretion ≥ 0.5 g/24 h in 24-h urine collection. Serum cholesterol was measured using standard laboratory procedures. Serum creatinine was assessed using a modified version of the Jaffé method (MEGA AU 510; Merck Diagnostica, Darmstadt, Germany). Erythrocytosis was defined as hemoglobin level higher than 16.0 g/dL for women, and higher than 16.5 g/dL for men [23].

### 2.4. Statistical Analyses 

Data were analyzed using IBM SPSS software, version 23.0 (SPSS Inc., Chicago, IL), R version 3.2.3 (Vienna, Austria) and STATA 14.1 (STATA Corp., College Station, TX). Data are expressed as mean ± standard deviation [SD] for normally distributed variables and as median (25th–75th interquartile range (IQR)) for variables with a skewed distribution. Categorical data are expressed as numbers (percentages). Co-linearity was tested by means of variance inflation factor (VIF) calculation, with a VIF score of lower than 5 indicating no evidence for co-linearity. We used Cox proportional hazards regression analysis to investigate the association between EPO levels and prospective outcomes. Assumptions of proportionality in Cox regression analyses were checked using Schoenfeld residuals plots and checking nonsignificance of covariates and with the global test (Model 1; EPO with death and CV death; *P* > 0.30 for global test). In these Cox regression analyses, we adjusted for potential confounders based on univariable associations or for factors of known biologic importance. We adjusted for age, sex, body surface area (BSA), eGFR, proteinuria, time since transplantation, presence of diabetes, systolic blood pressure (SBP), total cholesterol, and use of calcineurin inhibitors, proliferation inhibitors, and angiotensin-converting enzyme (ACE)-inhibitors and angiotensin II-receptor blockers (ARBs) (Model 1). We subsequently adjusted for potential mediators in the pathway between EPO and death, i.e., hemoglobin levels (Model 2); for ferritin (Model 3), high-sensitive C-reactive protein (hs-CRP) (Model 4), and finally for total FGF23 (Model 5). Due to skewed distribution, EPO, ferritin, hs-CRP, and total FGF23 were natural log-transformed. We repeated the Cox regression analyses between EPO and outcomes with EPO levels being divided in quartiles. Furthermore, we generated Kaplan–Meier curves to visually show the effect of increased risk of death and cardiovascular death while being in the highest EPO quartile. A log-rank test for trend was used to compare rates of death across quartiles. We also assessed the association between FGF23 levels and prospective outcomes adjusting for all potential confounders according to Model 1 and including EPO. To reflect the contribution of covariates in the different Cox regression models, we generated Appendix A showing the strength of covariates in univariable and multivariable models. To allow comparability between the hazard ratios (HR) of covariates, HR of continuous variables in Appendix A are shown as expressed per SD. As sensitivity analysis, we assessed the prevalence of different etiologies of CKD in total cohort and across EPO quartiles, and we adjusted the association between EPO and all-cause and cardiovascular death for etiology of CKD. Subsequently, we calculated the percentage of change in HR before and after adjustment for FGF23. Percentage change in HR was calculated as—(HR without adjustment – HR with adjustment)/(HR without adjustment – 1) × 100% [24]. Hereafter, we performed mediation analyses with the methods as previously described by Preacher and Hayes, which are based on logistic regression [25,26]. These analyses allow for testing significance and magnitude of mediation on the association between EPO and outcomes [25,26]. Overall, 0.4% of demographic data were missing and these data were imputed using regressive switching [27]. Five datasets were multiply-imputed, and results were pooled and analyzed according to Rubin’s rules [28]. In all analyses, a two-sided *p*-value < 0.05 was considered significant.

## 3. Results

### 3.1. Baseline Characteristics

We included 579 RTRs (mean age of 51 ± 12 years; 55% male) at a median of 6.0 (2.6–11.6) years after transplantation. Erythrocytosis was present in 27 (5%) of the included RTRs. Further demographics and clinical baseline characteristics across quartiles of EPO are shown in Table 1.

Median plasma EPO levels were 17.4 (11.9–24.2) IU/L and median FGF23 levels were 137 (94–212) RU/mL. Increased FGF23 levels were noted across EPO quartiles (115 (81–168) RU/mL; 125 (88–184) RU/mL; 138 (95–212) RU/mL; and 195 (115–363) RU/mL respectively, *P* < 0.001). FGF23 levels were positively correlated with EPO levels (r = 0.28, *P* < 0.001), with a VIF of 1.15, indicating very minimal co-linearity. 

### 3.2. EPO, FGF23, and Death

During a median follow-up of 7.0 (6.2–7.4) years, 121 RTRs died. Of the 121 deceased RTRs, 62 RTRs (51%) died from cardiovascular causes. Other causes of death were infection (18%), malignancy (24%), and miscellaneous causes (8%). 

In univariable Cox regression analyses, higher EPO levels were associated with an increased risk of all-cause death (HR per 1 ln IU/L increase, 1.74; 95% confidence interval (CI), 1.29–2.34; *P* < 0.001). A full list of HRs for covariates univariably with death are described in Appendix A. 

In multivariable Cox regression analyses, the association between EPO and all-cause death remained significant (HR, 1.66; 95% CI, 1.16-2.36; P=0.005) independent of adjustment for age, sex, BSA, eGFR, proteinuria, time since transplantation, presence of diabetes, SBP, total cholesterol, use of calcineurin inhibitors, proliferation inhibitors, ACE-inhibitors or ARB (Model 1). Further adjustment for hemoglobin, ferritin, or hs-CRP levels did not materially alter the results. However, further adjustment for FGF23 levels attenuated the association between EPO and all-cause death such that the association no longer remained significant (HR, 1.28; 95% CI, 0.87–1.88; *P* = 0.20) (Table 2). A full list of HRs for covariates in the multivariable model can be found in Appendix A.

We identified similar results when subdividing EPO levels into quartiles (Table 3; Figure 1 with Kaplan–Meier curves showing the univariably increased risk of death across EPO quartiles). 

In multivariable Cox regression analysis, RTRs in the upper quartile of EPO had a more than two times higher risk of death (HR, 2.11; 95% CI, 1.15–3.86), when compared to RTRs in the lowest quartile, independent of potential confounders. In line with the association between EPO as continuous variable and death, further adjustment for FGF23 levels attenuated the association between the upper EPO quartile and risk of death (HR, 1.55; 95% CI, 0.82–2.91; Table 3; Figure 2A). A full list of HRs for covariates in the multivariable model for EPO divided in quartiles can be found in Appendix A. 

When we assessed the associations between EPO and cardiovascular death, we found similar findings. Higher EPO levels were associated with an increased risk of cardiovascular death in univariable analyses (Figure 1) and in all subsequent models (Table 2). However, the association no longer remained significant, both as a continuous variable and as divided in quartiles, after further adjustment for FGF23 (Table 2 and Table 3; Figure 2B).

In contrast, FGF23 levels per se were strongly associated with all-cause death independent of adjustment for potential confounders including EPO (HR per RU/mL, 1.76; 95% CI, 1.33–2.34; *P* < 0.001). Likewise, FGF23 levels per se were also strongly associated with cardiovascular death independent of adjustment for potential confounders including EPO (HR, 1.84; 95% CI, 1.23–2.76; *P* = 0.003). A full list of HRs for covariates can be found in Appendix A. 

As sensitivity analysis, we assessed the prevalence of different etiologies of CKD in the total cohort and across quartiles of EPO (Appendix A). The most prevalent etiologies of CKD were primary glomerular disease (28%), polycystic disease (18%), and tubulo-interstitial disease (16%). Following adjustment for etiology of CKD additive to model 1, the association between EPO and all-cause death (HR, 1.57; 95% CI, 1.09–2.26; *P* = 0.02) and between EPO and cardiovascular death (HR, 1.75; 95% CI, 1.05–2.90; *P* = 0.03) remained materially unchanged. 

### 3.3. EPO, FGF23, and Graft Failure

During a median follow-up of 6.9 (6.1–7.4) years, 46 RTRs developed DCGF. When we assessed the associations between EPO and DCGF, we did not find an association (HR, 0.82; 95% CI, 0.48–1.41; *P* = 0.48). Further adjustment for potential confounders did not ameliorate the association between EPO and DCGF. In contrast, FGF23 levels were univariately associated with DCGF (HR, 3.07; 95% CI, 2.22–4.24; *P* < 0.001). However, after adjustment for potential confounders including EPO, FGF23 was no longer associated with DCGF (HR, 1.57; 95% CI, 0.94–2.64; *P* = 0.09).

### 3.4. Percentage Change HR and Mediation Analyses

Adjustment for FGF23 caused a large reduction in HR in the Cox Regression analysis in the association between EPO and all-cause and cardiovascular death (58% reduction in HR between EPO and all-cause death; and 48% reduction in HR between EPO and cardiovascular death). In subsequent mediation analyses, we identified that FGF23 was a significant mediator of the association between EPO and all-cause death (*P* value for indirect effect <0.05; 72% of the association was explained by FGF23; Table 4). Similarly, FGF23 explained 50% of the association between EPO and cardiovascular death (*P* value for indirect effect <0.05; Table 4).

## 4. Discussion

In this study, we show that higher endogenous EPO levels are associated with an increased risk of all-cause and cardiovascular death in RTRs, and that these associations are largely explained by variation in FGF23 levels. This study confirms recent studies about the essential role of EPO in FGF23 physiology in experimental and human models [7,8,9,10], extends these findings to RTRs, and support the notion that FGF23 is an important mediator in the association between EPO and risk of death. 

EPO, a hormone mainly produced in the kidney in response to hypoxia, is essential for erythropoiesis [29]. EPO controls proliferation, maturation, and also survival of erythroid progenitor cells [30]. Previously, it has been shown that high endogenous EPO levels were associated with an increased risk of all-cause and cardiovascular death in RTRs [2,3]. Similarly, in the setting of CKD and ESRD, correction of anemia with recombinant EPO led an increased risk of cardiovascular morbidity and death [5,6]. The mechanisms responsible for these adverse effects of both endogenous as exogenous EPO are unknown. In the Correction of Hemoglobin and Outcomes in Renal Insufficiency (CHOIR) trial, the highest risk of cardiovascular death was seen in patients with the highest EPO dose, suggesting that EPO resistance through inflammation and/or functional iron deficiency might be a possible link [6]. However, in the current study, the association between endogenous EPO levels and death was independent of adjustment for inflammation as well as independent of iron parameters, renal function, and standard classical cardiovascular risk factors including systolic blood pressure and cholesterol levels. Although there was a difference in prevalence of use of calcineurin and proliferation inhibitors across EPO quartiles, the association between EPO and death remained independent of adjustment for calcineurin and proliferation inhibitors. In contrast, adjustment for FGF23 markedly attenuated the association between EPO and death.

Elevated total FGF23 levels have previously been shown to be associated with increased risk of death in RTRs, as well as in various other patient groups including postoperative acute kidney injury, nondialysis CKD, and ESRD [31,32,33,34,35]. FGF23 regulation is determined by a complex interplay between parathyroid hormone, 1,25-dihydroxyvitamin D, klotho, glucocorticoids, calcium, and phosphate [36,37]. In recent years, iron deficiency has been identified as an important regulator of FGF23 [38,39,40]. In addition, recent studies demonstrated that EPO stimulates murine and human FGF23 [7,8]. Clinkenbeard and colleagues reported increased FGF23 mRNA expression in vitro, ex vivo, and in vivo due to EPO treatment in UMR-106 cells, in isolated bone marrow cells, and in marrow from mice, respectively [7]. In addition, Rabadi et al. showed in experimental animal models that an acute loss of 10% blood volume led to an increase in total FGF23 and EPO levels within six hours. Furthermore, exogenous administration of EPO resulted in an acute increase in plasma total FGF23 levels similar to those seen in acute blood loss [8]. Similarly, Flamme et al. described in animal models an increase in plasma total FGF23 both after injection of recombinant human EPO and after HIF-proline hydroxylase inhibitor [41]. The present findings in our study underscore these observations and emphasize the important role of EPO in FGF23 physiology. Importantly, the current study is the first to show that prospective associations between EPO and adverse outcomes in RTRs seem to be, at least to large extent, related to increased levels of total FGF23. 

The mechanisms through which EPO, as reflection of tissue hypoxia, lead to increased bone marrow FGF23 transcription are currently unknown and require additional investigation. The previously performed studies showed that EPO acutely increases total FGF23 levels out of proportion to intact FGF23 (iFGF23), suggesting an upregulated FGF23 production with concomitantly increased cleavage, with as a result an increase in C-terminal FGF23 fragments. To date, it remains incompletely understood how EPO increases post-translational cleavage. Results from our group and collaborators found previously in EPO-overexpressing mice a decreased GalNT3 bone marrow mRNA expression, without differences in Fam20C or furin expression, implying that a decreased GalNT3 might play a possible role [9]. However, more investigation is imperative to unravel this mechanistic link. 

The downstream consequences of elevated levels of FGF23 and the subsequent excess risk of death have not been fully elucidated yet. Several previous reports have shown that iFGF23 has biologic activity through binding to several FGF23 receptors. Besides the well-known functions of iFGF23 in the regulation of renal phosphate handling and vitamin D metabolism, recent studies have shown a myriad “off-target” effects of iFGF23 on the heart and other organs. Preclinical studies demonstrated that FGF23 can lead to left ventricular hypertrophy in cardiac myocytes, and promote endothelial dysfunction [42,43]. In addition, FGF23 stimulates renal fibrosis [44], exerts pro-inflammatory effects [45], and disrupts normal immune function [46]. Most likely, the increased death risk due to elevated levels of FGF23 is attributable to a combination of these effects. Although the biologic activity of iFGF23 has unequivocally been demonstrated, the biologic activity of C-terminal FGF23 fragments remains uncertain. Previously, it has been shown that C-terminal FGF23 may function as an iFGF23 antagonist, by competing with iFGF23 for binding to its receptor, which may reduce phosphaturia and aggravate soft tissue calcification [47]. In addition, Courbabaisse et al. has shown, at least in vitro, that C-terminal FGF23 increases adult rat ventricular cardiomyocyte size by stimulation of FGF receptor 4 in the absence of co-stimulatory factor alpha-klotho, and in sickle cell disease patients that elevated cleaved FGF23 levels were associated with heart hypertrophy [48].

Our study has multiple strengths as well as limitations. The major strength of the current study is the large prospective cohort of stable RTRs with detailed clinical and laboratory data available, including EPO, FGF23, hs-CRP, and ferritin levels. Additionally, no participants were lost to follow-up with respect to the endpoints, despite a considerable follow-up period. Limitations of the current study include that, due to the observational status of our single center study, we cannot exclude the possibility of residual confounding, and conclusions about causality cannot be drawn. Furthermore, we were unable to measure iFGF23 levels, since samples were not stored with protease inhibitors, and iFGF23 has been shown to be susceptible to degradation with long-term storage [49]. This precludes us to discern whether the elevated total FGF23 levels are the result of increased iFGF23 levels or due to an increase in allegedly assumed inactive C-terminal fragments. Another limitation of the study is that we only used CRP levels as inflammatory parameter, other markers of inflammation (e.g., cytokines, cell subtypes) were not available, but could possibly have contributed to the results. In addition, another limitation of current study is the use of single-time measurements hampering the possibility to track the levels of EPO and FGF23 over time with respect to each other and with respect to risk of death. However, it should be realized that most epidemiological studies use a single baseline measurement to investigate associations of variables with outcomes, which adversely affects the strength and significance of the association of these variables with outcomes. If intraindividual variability of variables is taken into account, this results in strengthening of associations that also existed for single measurements of these variables [50,51]. Finally, we want to emphasize that the mediation analyses that we performed are plain straightforward mediation analyses. Although based on literature, we have strong evidence that FGF23 is a mediator in the association of EPO with risk of death, we cannot exclude that an unmeasured cause of mortality or alternative potential mediators has influenced the currently identified results.

In conclusion, we identified that elevated levels of EPO were independently associated with an increased risk of death in RTRs, and that this association was to a large extent explained by variation in FGF23 levels. Further research is needed to fully elucidate the mechanism through which this ensues and to unravel whether the currently identified results can be extrapolated to exogenous EPO in RTRs.

## Figures and Tables

**Figure 1 jcm-09-01737-f001:**
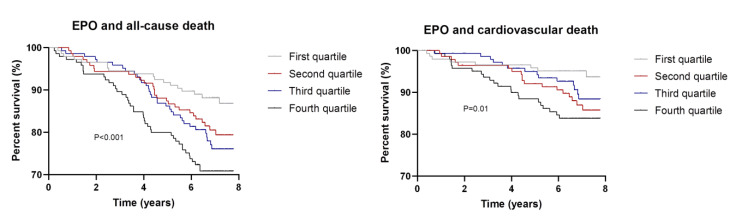
Kaplan–Meier Curves depicting the association between EPO quartiles and risk of all-cause (left panel) and cardiovascular death (right panel). Reported p-values have been calculated with the log-rank test for trend.

**Figure 2 jcm-09-01737-f002:**
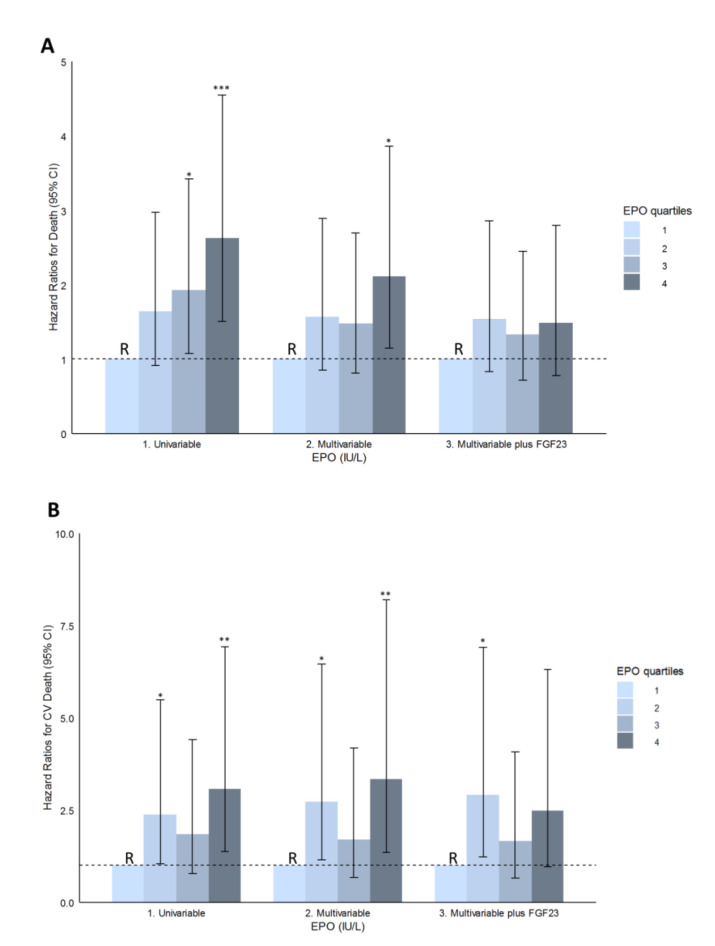
Hazard ratios and corresponding 95% confidence intervals are depicted for risk of death, both all-cause (**A**) and cardiovascular death (**B**), according to quartiles of erythropoietin levels. First, the univariate association is shown. Second, the multivariable adjustment is performed with adjustment for age, sex, body surface area (BSA), eGFR, proteinuria, time since transplantation, presence of diabetes, systolic blood pressure (SBP), total cholesterol, and use of calcineurin inhibitors, proliferation inhibitors, and angiotensin-converting enzyme (ACE)-inhibitors and angiotensin II-receptor blockers (ARBs). Third, adjustment for FGF23 is performed following the multivariable adjustment. The first quartile was chosen as a reference group in all analyses. Significance levels are indicated by numbers of asterisks, i.e., *** <0.001, ** <0.01, * <0.05; Abbreviations—BSA, body surface area; CI, confidence interval; eGFR, estimated glomerular filtration rate.

**Table 1 jcm-09-01737-t001:** Baseline characteristics of the included 579 renal transplant recipients (RTRs) across erythropoietin (EPO) quartiles.

		Quartiles of EPO (IU/L)				
	All Patients (*n* = 579)	Q1 (*n* = 146) [4.0–11.9]	Q2 (*n* = 143) [12.0–17.2]	Q33 (*n* = 145) [17.3–23.9]	Q4 (*n* = 145) [24.2–182.0]	*P*-value
Age (years)	51 ± 12	47 ± 13	50 ± 12	53 ± 11	54 ± 11	<0.001
Male sex (*n*, %)	317 (55)	92 (63)	78 (55)	73 (50)	74 (51)	0.11
Body surface area, m^2^	1.87 ± 0.19	1.86 ± 0.19	1.86 ± 0.20	1.87 ± 0.18	1.88 ± 0.20	0.81
Alcohol use (*n*, %)	290 (50)	77 (52)	73 (51)	64 (44)	76 (52)	0.35
Smoking status						0.29
Never smoker (*n*, %)	205 (35)	47 (32)	49 (34)	59 (41)	50 (35)	
Former smoker (*n*, %)	246 (43)	64 (44)	68 (48)	59 (41)	55 (38)	
Current smoker (*n*, %)	126 (22)	35 (24)	25 (18)	27 (19)	39 (27)	
Time since Tx (yrs)	6.0 (2.6–11.6)	4.6 (2.1–9.2)	5.7 (3.1–11.2)	6.5 (3.3–12.4)	7.0 (2.8–13.7)	0.007
Diabetes mellitus (*n*, %)	102 (18)	25 (17)	21 (15)	27 (19)	29 (20)	0.67
SBP (mmHg)	153 ± 23	150 ± 20	151 ± 21	153 ± 22	157 ± 26	0.05
DBP (mmHg)	90 ± 10	90 ± 9	90 ± 9	90 ± 10	90 ± 11	0.91
Laboratory measurements						
FGF23 (RU/mL)	137 (94–212)	115 (81–168)	125 (88–184)	138 (95–212)	195 (115–363)	<0.001
Hemoglobin (g/dL)	13.9 ± 1.5	14.2 ± 1.5	14.0 ± 1.5	13.8 ± 1.5	13.5 ± 1.5	0.001
Erythrocytosis (*n*, %)^‡^	27 (5)	11 (8)	8 (6)	5 (3)	3 (2)	0.13
MCV (fL)	91 ± 6	89 ± 4	91 ± 6	92 ± 6	92 ± 8	<0.001
Ferritin (µg/L)	154 (76–282)	154 (76–320)	164 (100–305)	159 (89–283)	118 (61–240)	0.02
Total cholesterol (mmol/L)	5.6 ± 1.1	5.7 ± 0.9	5.7 ± 1.3	5.6 ± 1.0	5.6 ± 1.1	0.78
Phosphate (mmol/L)	1.1 ± 0.2	1.05 ± 0.21	1.07 ± 0.21	1.05 ± 0.19	1.07 ± 0.22	0.85
Calcium (mmol/L)	2.39 ± 0.16	2.39 ± 0.14	2.38 ± 0.16	2.39 ± 0.18	2.41 ± 0.15	0.51
Vit. 25(OH) D, nmol/l *	53 ± 23	52 ± 24	51 ± 21	57 ± 21	53 ± 25	0.33
Vit. 1,25(OH)_2_ D, pmol/L*	109 ± 46	106 ± 50	112 ± 46	107 ± 40	110 ± 47	0.80
PTH (pmol/L)	9.1 (6.0–13.4)	8.8 (6.2–13.2)	9.6 (5.8–13.9)	9.2 (6.0–13.7)	8.9 (6.0–14.0)	0.93
eGFR (ml/min/1.73m^2^)	48 ± 15	50 ± 16	48 ± 14	47 ± 15	46 ± 16	0.16
Creatinine (µmol/L)	144 ± 52	145 ± 51	139 ± 40	142 ± 51	148 ± 62	0.50
Proteinuria (>0.5g) (*n*, %)	155 (27)	34 (23)	32 (22)	39 (27)	50 (35)	0.07
hs-CRP (mg/L)	2.0 (0.8–4.8)	1.4 (0.6–3.8)	2.0 (0.7–4.1)	2.1 (1.0–4.2)	3.2 (1.2–7.2)	<0.001
Treatment						
ACE-i/AII-antagonists (*n*, %)	190 (33)	65 (45)	50 (35)	37 (26)	38 (26)	0.001
ACE-I (*n*, %)	154 (27)	53 (36)	37 (26)	31 (24)	33 (23)	
AII-antagonists (*n*,%)	36 (6)	12 (8)	13 (9)	6 (4)	5 (3)	
Bèta-blocker (*n*, %)	356 (62)	88 (63)	96 (67)	81 (56)	91 (63)	0.26
Ca^2+^ channel blockers (*n*, %)	220 (39)	56 (38)	52 (36)	51 (35)	61 (42)	0.45
Diuretic use (*n*, %)	250 (43)	52 (36)	62 (43)	60 (41)	76 (52)	0.04
Proliferation inhibitor (*n*, %)	428 (74)	95 (65)	99 (69)	115 (79)	119 (82)	0.002
Azathioprine (*n*, %)	187 (32)	18 (12)	42 (29)	55 (38)	72 (50)	
Mycophenolic acid (*n*, %)	241 (42)	77 (53)	57 (40)	60 (41)	47 (32)	
Calcineurin inhibitor (*n*, %)	457 (79)	131 (90)	120 (84)	103 (71)	103 (71)	<0.001
Ciclosporin (*n*,%)	376 (65)	108 (78)	101 (71)	81 (56)	86 (59)	
Tacrolimus (*n*, %)	81 (14)	23 (16)	19 (13)	22 (15)	17 (12)	

^‡^ Erythrocytosis defined as hemoglobin level >16 g/dL (F) and >16.5 g/dL (M) * Only available in a subset cohort of 415 RTRs. Values are means ± standard deviation, medians (interquartile range) or proportions (%). Diabetes mellitus was defined as serum glucose > 7 mmol/L or the use of antidiabetic drugs. Abbreviations—ACE-i, angiotensin converting enzyme inhibitors; DBP, diastolic blood pressure; eGFR, estimated glomerular filtration rate; FGF23, fibroblast growth factor 23; hs-CRP, high-sensitivity C-reactive protein; MCV, mean corpuscular volume; SBP, systolic blood pressure; Tx, transplantation.

**Table 2 jcm-09-01737-t002:** Association between erythropoietin levels and risk of all-cause and cardiovascular death.

	EPO (IU/L)	
**All–cause death**	**HR (95% CI) ***	***P*-value**
Univariable	1.74 (1.29–2.34)	<0.001
Model 1	1.66 (1.16–2.36)	0.005
Model 2	1.72 (1.21–2.46)	0.003
Model 3	1.80 (1.25–2.60)	0.002
Model 4	1.60 (1.12–2.29)	0.01
Model 5	1.28 (0.87–1.88)	0.20
**Cardiovascular death**	**HR (95% CI) ***	***P*-value**
Univariable	1.70 (1.12–2.58)	0.01
Model 1	1.87 (1.14–3.06)	0.01
Model 2	1.90 (1.16–3.12)	0.01
Model 3	2.05 (1.22–3.44)	0.006
Model 4	1.87 (1.14–3.06)	0.01
Model 5	1.45 (0.84–2.48)	0.18

* Hazard ratios are shown per 1 ln IU/L increase in EPO levels; Model 1: Adjusted for age, sex, body surface area, eGFR, proteinuria, time since transplantation, presence of diabetes, systolic blood pressure, total cholesterol, use of calcineurin inhibitors, proliferation inhibitors, and ACE-inhibitors or ARB; Model 2: Model 1 + adjustment for hemoglobin; Model 3: Model 1 + adjustment for ferritin; Model 4: Model 1 + adjustment for hs-CRP; Model 5: Model 1 + adjustment for FGF23. Ferritin, hs-CRP, and FGF23 were naturally log transformed before adding to the Cox regression analysis due to skewed distribution. Abbreviations—ACE, angiotensin-converting enzyme; ARB, angiotensin-receptor blockers; FGF23, fibroblast growth factor 23; CI, confidence interval; eGFR, estimated glomerular filtration rate; HR, hazard ratio; hs-CRP, high-sensitive C-reactive protein.

**Table 3 jcm-09-01737-t003:** Association between erythropoietin quartiles and risk of all-cause and cardiovascular death.

	Quartiles of EPO (IU/L)			
	Q1	Q2	Q3	Q4
**All–cause death**	**Ref**	**HR (95% CI)**	**HR (95% CI)**	**HR (95% CI)**
Univariable	1.00	1.64 (0.91–2.97)	1.93 (1.08–3.42)	2.62 (1.51–4.55)
Model 1	1.00	1.57 (0.85–2.89)	1.47 (0.81–2.69)	2.11 (1.15–3.86)
Model 2	1.00	1.54 (0.83–2.84)	1.51 (0.83–2.75)	2.19 (1.19–4.05)
Model 3	1.00	1.65 (0.88–3.11)	1.57 (0.84–2.93)	2.29 (1.21–4.31)
Model 4	1.00	1.55 (0.85–2.85)	1.42 (0.78–2.58)	1.99 (1.09–3.65)
Model 5	1.00	1.63 (0.89–3.01)	1.41 (0.77–2.57)	1.55 (0.82–2.91)
**Cardiovascular Death**	**Ref**	**HR (95% CI)**	**HR (95% CI)**	**HR (95% CI)**
Univariable	1.00	2.38 (1.04–5.48)	1.85 (0.78–4.40)	3.08 (1.37–6.92)
Model 1	1.00	2.73 (1.15–6.45)	1.69 (0.68–4.18)	3.34 (1.36–8.20)
Model 2	1.00	2.58 (1.08–6.16)	1.71 (0.69–4.25)	3.41 (1.31–8.47)
Model 3	1.00	2.91 (1.17–7.23)	1.77 (0.68–4.59)	3.57 (1.39–9.20)
Model 4	1.00	2.65 (1.12–6.25)	1.60 (0.65–3.95)	3.10 (1.27–7.60)
Model 5	1.00	2.90 (1.22–6.91)	1.65 (0.66–4.08)	2.47 (0.97–6.31)

Model 1: Adjusted for age, sex, body surface area, eGFR, proteinuria, time since transplantation, presence of diabetes, systolic blood pressure, total cholesterol, use of calcineurin inhibitors, proliferation inhibitors, and ACE-inhibitors or ARB. Model 2: Model 1 + adjustment for hemoglobin; Model 3: Model 1 + adjustment for ferritin; Model 4: Model 1 + adjustment for hs-CRP; Model 5: Model 1 + adjustment for FGF23. Ferritin, hs-CRP, and FGF23 were naturally log transformed before adding to the Cox regression analysis due to skewed distribution. Abbreviations—ACE, angiotensin-converting enzyme; ARB, angiotensin-receptor blockers; FGF23, fibroblast growth factor 23; CI, confidence interval; eGFR, estimated glomerular filtration rate; HR, hazard ratio; hs-CRP, high-sensitive C-reactive protein.

**Table 4 jcm-09-01737-t004:** Mediation analysis of FGF23 on the association between EPO and all-cause and cardiovascular death in renal transplant recipients.

Potential Mediator	Outcome	Effect (path) *	Multivariable Model **	
			Coefficient (95% CI, bc) †	Proportion Mediated ***
FGF23	All-cause death	Indirect effect (ab path)	0.090 (0.044; 0.139)	72%
		Total effect (ab + c’ path)	0.124 (−0.011; 0.255)	
		Unstandardized total effect ‡	0.120 (−0.385; 0.624)	
FGF23	Cardiovascular death	Indirect effect (ab path)	0.065 (0.015; 0.122)	50%
		Total effect (ab + c’ path)	0.129 (−0.040; 0.290)	
		Unstandardized total effect ‡	0.218 (−0.405; 0.840)	

* The coefficients of the indirect *ab* path and the total *ab* + *c’* path are standardized for the standard deviations of EPO, FGF23, all-cause and cardiovascular death. **All coefficients are adjusted for age, sex, body surface area, eGFR, proteinuria, time since transplantation, presence of diabetes, systolic blood pressure, total cholesterol, use of calcineurin inhibitors, proliferation inhibitors, ACE-inhibitors or ARB. *** The size of the significant mediated effect is calculated as the standardized indirect effect divided by the standardized total effect multiplied by 100, e.g., 0.090 divided by 0.124 multiplied by 100 constitutes 72% as percentage of mediation ‡ Odds ratios for risk of outcomes can be calculated by taking the exponent of the unstandardized total effect. For example, the unstandardized coefficient of the direct effect of EPO on all-cause death while adjusting for FGF23 is 0.120, which can be calculated to an OR by taking the exponent of this regression coefficient, i.e., e^0.120^=1.12, which corresponds to the HR of 1.28 (see Table 2). The discrepancy between the ratios is due to taking into account time-to-event with HR in contrast to OR. †95% CIs for the indirect and total effects were bias-corrected confidence intervals after running 2000 bootstrap samples. Abbreviations—ACE, angiotensin converting enzyme; ARB, angiotensin receptor blockers; Bc, bias corrected; CI, confidence interval; eGFR, estimated glomerular filtration rate; FGF23, fibroblast growth factor 23.

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
