# Peer review of "Erythropoietin, Fibroblast Growth Factor 23, and Death After Kidney Transplantation"

_jcm, 2020, doi:10.3390/jcm9061737_

Round 1

Reviewer 1 Report

In my opinion the paper is well written.

I have only 2 questions

  1. Did the authors analyse the post KTx EPO levels in the context of the etiology of CKD? In some patients the native kidneys may be the source of EPO. Maybe an additional table?
  2. 4 or 5 patients I take care of in the outpatient clinic have clinically significant polycytemia and need a phlebotomy procedure 2-3 times a year despite the ACE-I/ARB administration. Did the authors have such patients in the analyses?

Author Response

In my opinion the paper is well written.

Response: We thank the reviewer for the kind words.

I have only 2 questions

  1. Did the authors analyse the post KTx EPO levels in the context of the etiology of CKD? In some patients the native kidneys may be the source of EPO. Maybe an additional table?

We thank the reviewer for the comment. We agree with the reviewer that data on the etiology of CKD are of added value for current manuscript. To accommodate the comment of the reviewer, we have made an additional Table (Supplemental Table 5) in which we describe the prevalence of the different etiologies of CKD in the total cohort and across EPO quartiles. Of the different etiologies of CKD, primary glomerular disease (28%), polycystic disease (18%), and tubulo-interstitial disease (16%) were the most prevalent in the included RTRs. Across quartiles of EPO, the prevalence of four etiologies of CKD, i.e. primary glomerular disease, glomerulonefritis, tubulo-interstitial disease and polycystic kidney disease, were significantly different, with in the highest quartile of EPO prevalence of polycystic kidney disease being relatively high, and of primary glomerular disease and glomerulonephritis being relatively low. In general, the etiology of CKD was significantly different across EPO quartiles (p=0.005). Hence, as sensitivity analysis, we adjusted the association between EPO and all-cause and cardiovascular death for etiology of CKD. The association between EPO and all-cause death (HR, 1.57; 95% CI, 1.09-2.26; P=0.02) and between EPO and cardiovascular death (HR, 1.75; 95% CI, 1.05-2.90; P=0.03) remained materially unchanged following adjustment for etiology of CKD additive to model 1. We have added the information about the Supplemental Table 5 and the sensitivity analysis to the revised version of the manuscript (lines 134-137 of the methods section of the revised manuscript and lines 219-224 of the results section of the revised manuscript).

  1. 4 or 5 patients I take care of in the outpatient clinic have clinically significant polycytemia and need a phlebotomy procedure 2-3 times a year despite the ACE-I/ARB administration. Did the authors have such patients in the analyses?

Response: We thank the reviewer for the comment for bringing up this point. Accordingly, we have assessed in our observational cohort the presence of erythrocytosis, as defined by a hemoglobin level higher than 16 g/dL for women, and higher than 16.5 g/dL for men (Arber DA et al. Blood 2016). In our cohort, we identified erythrocytosis in 27 (5%) of the included RTRs. Hereafter, we assessed whether the prevalence of erythrocytosis was different across the quartiles of EPO, but this was not the case (p=0.13). We have added the information about the prevalence of erythrocytosis to the revised version of the manuscript (line 103-105 of the methods section; and line 147 of the results section) and we added the prevalence of erythrocytosis across quartiles of EPO also to the revised Table 1.

Reviewer 2 Report

In the present paper authors described the correlation between EPO, FGF23 levels and increased risk of death in renal transplanted recipients. These evidences have been already described in literature. The presented data are limited to a single cohort and have not been confirmed in an independent cohort. Moreover, also methodological analysis of FGF23 presents limitations since it does not consider iFGF23 levels. 

Results reported in the present manuscript have been already described. The paper present several methodological limitations.

Author Response

In the present paper authors described the correlation between EPO, FGF23 levels and increased risk of death in renal transplanted recipients. These evidences have been already described in literature. The presented data are limited to a single cohort and have not been confirmed in an independent cohort. Moreover, also methodological analysis of FGF23 presents limitations since it does not consider iFGF23 levels. 

Results reported in the present manuscript have been already described. The paper present several methodological limitations.

Response: We regret that the reviewer surmises that the presented results have already been described. This is however not the case, since the results within this manuscript are the first to highlight a possible important role for variation in FGF23 levels with respect to EPO and its associated increased risk of death in the patient setting of RTRs. We agree with the reviewer that the manuscript has limitations which we have been mentioned in lines 349-368 of the manuscript. Future studies are necessary to further elaborate on the currently identified findings in our observational cohort.

Reviewer 3 Report

The article deals with the influence of FGF23 levels on the survival of patients who have undergone a kidney transplant.
Although the elevation of EPO is a risk factor for death in kidney transplant recipients (the mechanism of action is unknown), the authors suggest that this effect would be measured by the elevation of FGF23 induced by EPO.
The paper may provide information to better understand these mechanisms, but it needs a major review before it can be accepted.

Material and methods.
Although it is said that patients with more than one year of survival after transplantation were recruited. The exact time at which the blood samples were taken is not indicated. It is also not indicated whether further studies were conducted to determine if those elevated levels were maintained over time. It would be interesting to be informed if the markers are tracked over time. It would be of great interest to assess whether the evolution of EPO and GFG23 levels in patients has any kind of association with the evolution and survival of patients.
There is talk of death of cardiovascular origin, but nevertheless it is not described what has been the definition that has been used in the work to consider this rating. This definition should be incorporated.
The statistics section is convoluted, it should be described in a way that is easily understood by readers. Readers' understanding of mediation analyzes should be facilitated by incorporating rules for quick interpretation of the coefficients (rules of thumb).

Results
The text is hard, difficult to assimilate. The data is communicated with little clarity, too complicated. Greater simplicity will translate into better readability.
The median years after transplantation described in methods (page 2 line 71) contrasts with the median described at the beginning of the results: 6.0 (2.6-11.6) (line 122). Later on line 139, the method data is returned. This way of communicating the median is confusing. Is this the median of the total group? Of the excluded patients ?, of the included patients? Change the reference point to calculate the median ?. References should always be better explained and always made to the same point in time.Table 1. To analyze the different drugs that these patients have taken, grouped by ACEI, anti calcineurinics and anti proliferative agents, however, the latter do not define them, it should be clarified if they refer to mTOR inhibitors, mycophenolate,…. , it should be clarified to make easier to read and understand the treatment.

If they analyze death over time, survival graphs based on variables and tables of patients at risk (Kapplan Meier analysis) should be included in order to make reading comprehension much easier. The images would greatly facilitate the readability of a text that is difficult to assimilate due to the low friendliness of the tables.
Table 2. It is very confusing and does not reflect the contribution of the rest of the covariates in the respective models. A table describing the HR and the corresponding 95% CI of all the variables incorporated in the different multivariable analysis models is missing. This table should also indicate the univariable HR value of all of them.
Since the incorporation of FGF23 into the model implies the loss of the significance of erythropoietin, a table should be included comparing HR of FGF23 with HR of the rest of the covariates (excluding erythropoietin) to clearly assess the statistical strength of FGF23.
Table 3. should also be made more readable by incorporating the HR values for all covariates in the different models.
Table 4. A rule of thums should be provided for the interpretation of the coefficients in the information in the table itself or in the method statistics section.

Discussion:
Regarding the drugs used in patients, the authors do not discuss why there are significant differences between the number of patients who have taken them in the different quartiles. It seems that there are fewer patients treated with anticalcineurinics in Q4 patients (that is, those with more EPO levels), so it could be debated whether it would be a drug of choice for this type of patient. This aspects must be discussed.
Despite treating the death of patients as the main object of analysis, it is striking that at no time the authors analyze graft losses (graft survival). Does elevation of EPO or FGF23 have any relevance in renal graft loss in this cohort? Although there is no significance, it should be reflected in the manuscript, after all, the selection criteria on the study is the kidney transplant.
The possible mechanisms that mediate the adverse effect of FGF23 in the patients in this study are not discussed. Although references from other preclinical studies are provided on the effect of FGF 23 on ventricular hypertrophy, renal fibrosis, proinflammatory effects and endothelial dysfunction.
This work can be an excellent opportunity to describe whether or not there is an association in transplant patients and shed more light on these mechanisms. Providing this information should not require much effort since, as the authors say in the section on strengths, they have detailed clinical and laboratory data.

Author Response

The article deals with the influence of FGF23 levels on the survival of patients who have undergone a kidney transplant.
Although the elevation of EPO is a risk factor for death in kidney transplant recipients (the mechanism of action is unknown), the authors suggest that this effect would be measured by the elevation of FGF23 induced by EPO.
The paper may provide information to better understand these mechanisms, but it needs a major review before it can be accepted.

Material and methods.
Although it is said that patients with more than one year of survival after transplantation were recruited. The exact time at which the blood samples were taken is not indicated. It is also not indicated whether further studies were conducted to determine if those elevated levels were maintained over time. It would be interesting to be informed if the markers are tracked over time. It would be of great interest to assess whether the evolution of EPO and GFG23 levels in patients has any kind of association with the evolution and survival of patients.

Response: We thank the reviewer for the comment, and stand corrected for not having been sufficiently clear in the original manuscript. The included renal transplant recipients are from the outpatient follow up from patients who have been transplanted in our University Medical Center Groningen. The outpatient follow up constitutes a continuous surveillance system in which patients visit the outpatient clinic in our hospital with declining frequency, ranging from twice a week immediately after hospital discharge following the renal transplantation, to once a year long term after transplantation. For current observational study, stable renal transplant recipients were included (having a functioning graft for more than 1 year post transplantation). After written informed consent from the patients, blood samples were taken from these renal transplant recipients during their next scheduled outpatient clinic visit. We have added this information to the revised version of the manuscript (line 88 of the methods section). We have no repeated measurements in this cohort, and hence it is not possible to track the markers over time. We agree that this is a limitation of our manuscript. It should, however, be realized that most epidemiological studies use a single baseline measurement for studying the association of variables with outcomes, which adversely affects the strength and significance of the association of these variables with outcomes. If intra-individual variability of variables is taken into account, this results in strengthening of associations that – despite sometimes considerable intra-individual day-to-day variation – also existed for single measurements of these variables (Koenig W et al. Am J Epidemiol 2003; 158: 357-64, Danesh J et al. N Engl J Med 2004; 350: 1387-97). The higher the intra-individual day-to-day variation, the greater one would expect the benefit of inclusion of repeated measurements for assessment of the association with outcomes (Koenig W et al. Am J Epidemiol 2003; 158: 357-64, Danesh J et al. N Engl J Med 2004; 350: 1387-97). Thus, our use of a single measurement rather than several or multiple ones, provides an underestimation of an otherwise existing true effect. To accommodate the comment of the reviewer, we have added the limitation of a single time measurement hampering to track levels over time, but also stated that most likely it involves an underestimation of the true effect (lines 357-364 of the discussion section).

There is talk of death of cardiovascular origin, but nevertheless it is not described what has been the definition that has been used in the work to consider this rating. This definition should be incorporated.

Response: We thank the reviewer for the comment. The cause of death was obtained as coded by a physician from the Central Bureau of Statistics according to the International Classification of Diseases, 9th revision (ICD-9). For cardiovascular death, this ranges reporting of ICD-codes 410 to 447. The full list of diagnoses at time of death which has been considered cardiovascular in nature can be found at https://icd.codes/icd9cm. In the revised version of the manuscript, we have added specifically that the International Classification of Diseases was utilized to ascertain a cardiovascular cause of death and we included the link to the website which mentions all ICD-9 codes (lines 66-70 of the methods section of the revised version of the manuscript).

The statistics section is convoluted, it should be described in a way that is easily understood by readers. Readers' understanding of mediation analyzes should be facilitated by incorporating rules for quick interpretation of the coefficients (rules of thumb).

Response: We thank the reviewer for the comment. We increased simplicity of the text to allow better readability of the statistical section in the revised version of the manuscript. In addition, as requested by the reviewer, we have added to the mediation Table an explanation how the coefficients can be quickly interpreted. The size of the significant mediated effect is calculated as the standardized indirect effect divided by the standardized total effect multiplied by 100, e.g. 0.090 divided by 0.124 multiplied by 100 constitutes 72% as percentage of mediation. We have added this information and the example to the revised version of the manuscript (lines 270-272 of the results section).

Results
The text is hard, difficult to assimilate. The data is communicated with little clarity, too complicated. Greater simplicity will translate into better readability.

Response: We thank the reviewer for noting this. As requested by the reviewer, we increased simplicity of the text to allow better readability of the whole results section.

The median years after transplantation described in methods (page 2 line 71) contrasts with the median described at the beginning of the results: 6.0 (2.6-11.6) (line 122). Later on line 139, the method data is returned. This way of communicating the median is confusing. Is this the median of the total group? Of the excluded patients ?, of the included patients? Change the reference point to calculate the median ?. References should always be better explained and always made to the same point in time.

Response: We thank the reviewer for the comment and stand corrected for the unclarity. The median 7.0 (interquartile range, 6.2 to 7.4) years described in the methods section (page 2 line 77) and results section (page 5 line 164) corresponds to the median follow-up time. This is the follow-up time starting from the inclusion of the study (baseline) to endpoint (i.e. death or end of follow-up). This is in contrast to the time period described in line 146, which is the median time after transplantation which is 6.0 (interquartile range, 2.6-11.6) years (line 146). This is the median numbers of years after transplantation at which the RTRs were included in current study. To improve clarity, we have tried to make this more clear in the revised version of the manuscript (line 76 of the methods section).

Table 1. To analyze the different drugs that these patients have taken, grouped by ACEI, anti calcineurinics and anti proliferative agents, however, the latter do not define them, it should be clarified if they refer to mTOR inhibitors, mycophenolate,…. , it should be clarified to make easier to read and understand the treatment.

Response: As requested by the reviewer, we splitted the combined group of ACEi and AII-antagonists into separate reporting of ACEi and AII-antagonists, and we added the subdivision of the calcineurin inhibitors and proliferation inhibitors. With calcineurin inhibitors, we provided the subdivision into ciclosporin or tacrolimus which are the default calcineurin inhibitors used here in the Netherlands. Similarly, we subdivided the proliferation inhibitors into azathioprine and mycophenolate mofetil. We added this information about the subdivided forms of the ACEi, calcineurin inhibitors and proliferation inhibitors to the revised version of Table 1 in the revised manuscript.

If they analyze death over time, survival graphs based on variables and tables of patients at risk (Kapplan Meier analysis) should be included in order to make reading comprehension much easier. The images would greatly facilitate the readability of a text that is difficult to assimilate due to the low friendliness of the tables.

Response: We agree with the reviewer that survival graphs based on variables and tables of patients at risk (Kaplan Meier analysis) increases reading comprehension among readers. As requested by the reviewer, we have added Kaplan Meier analysis according to quartiles of EPO with respect to risk of all-cause death and cardiovascular death to the revised version of the manuscript (Figure 1 in the revised version of the manuscript; lines 127-129 of the methods section). In addition, we have also added to the revised version of the manuscript that assumptions of proportionality were not violated as assessed with non-significance of the Schoenfeld residuals and the global test. We have also added this information to the revised version of the manuscript (lines 115-117 of the methods section).

Table 2. It is very confusing and does not reflect the contribution of the rest of the covariates in the respective models. A table describing the HR and the corresponding 95% CI of all the variables incorporated in the different multivariable analysis models is missing. This table should also indicate the univariable HR value of all of them.
Since the incorporation of FGF23 into the model implies the loss of the significance of erythropoietin, a table should be included comparing HR of FGF23 with HR of the rest of the covariates (excluding erythropoietin) to clearly assess the statistical strength of FGF23.

Response: We thank the reviewer for the comment. To accommodate the comment of the reviewer, we have added as Supplemental Table 1 a Table with all univariable HR and corresponding 95% of the included variables which we have used later in the multivariable model on both outcomes of death and cardiovascular death. In addition, we have added as Supplemental Table 2 the HRs of all covariates in the multivariable analysis model. Furthermore, as requested by the reviewer, we have also added a Supplemental Table 4 involving the multivariable model of FGF23 (excluding erythropoietin) to assess the statistical strength of FGF23. To allow the reviewer better comparability between the different HR with different units, the HR of the continuous variables in the Supplemental Tables are expressed as Z-scores (per SD). We have mentioned generation of the Supplemental Tables to the methods section (lines 131-134 of the methods section) and included the Supplemental Tables 1,2 and 4 in the results section of the revised manuscript (lines 169-170; lines 188-189; and line 218 of the results section).

Table 3. should also be made more readable by incorporating the HR values for all covariates in the different models.

Response: Accordingly, as requested by the reviewer, we have also added this Table with all HR from covariates as Supplemental Table 3. We have also added mentioning of Supplemental Table 3 in the revised version of the manuscript (lines 206-207 of the results section).

Table 4. A rule of thums should be provided for the interpretation of the coefficients in the information in the table itself or in the method statistics section.

Response: As requested by the reviewer, we have added to the mediation Table an explanation how the coefficients can be quickly interpreted. The size of the significant mediated effect is calculated as the standardized indirect effect divided by the standardized total effect multiplied by 100, e.g. 0.090 divided by 0.124 multiplied by 100 constitutes 72% as percentage of mediation. We have added this information and the example to the revised version of the manuscript (lines 270-272 of the results section).

Discussion:
Regarding the drugs used in patients, the authors do not discuss why there are significant differences between the number of patients who have taken them in the different quartiles. It seems that there are fewer patients treated with anticalcineurinics in Q4 patients (that is, those with more EPO levels), so it could be debated whether it would be a drug of choice for this type of patient. This aspects must be discussed.

Response: We thank the reviewer for the comment. We agree with the reviewer that there are fewer patients using calcinurin inhibitors in the upper quartile of EPO. We would like to emphasize that we are adjusting in all analyses for use of calcinurin inhibitors and as such we try to account for a potential difference in use of calcinurin inhibitors across quartiles of EPO. In all analyses, the associations remain independent of adjustment for use of calcinurin inhibitors and proliferation inhibitors. To accommodate the comment of the reviewer, we have added that although the prevalence of use of calcinurin and proliferation inhibitors is different across erythropoietin quartiles, the association between erythropoietin and death remained (lines 296 – 299 of the discussion section).

Despite treating the death of patients as the main object of analysis, it is striking that at no time the authors analyze graft losses (graft survival). Does elevation of EPO or FGF23 have any relevance in renal graft loss in this cohort? Although there is no significance, it should be reflected in the manuscript, after all, the selection criteria on the study is the kidney transplant.

Response: We understand the comment of the reviewer. Therefore, to accommodate the comment of the reviewer, we have also assessed the development of graft survival. We have added the association between EPO and risk of death-censored graft failure (DCGF) to the revised version of the manuscript. Similarly, we have added the association between FGF23 and risk of DCGF to the revised version of the manuscript. During a median follow-up of 6.9 (6.1 – 7.4) years, 46 RTRs developed DCGF. When we assessed the associations between EPO and DCGF, we did not find an association (HR, 0.82; 95% CI, 0.48-1.41; P=0.48). Further adjustment for potential confounders did not ameliorate the association between EPO and DCGF. In contrast, FGF23 levels were univariately associated with DCGF (HR, 3.07; 95% CI, 2.22-4.24; P<0.001). However, after adjustment for potential confounders including EPO, FGF23 was no longer associated with DCGF (HR, 1.57; 95% CI, 0.94-2.64; P=0.09). In sum, both EPO and FGF23 levels were not associated with risk of death-censored graft failure over time following adjustment for potential confounders. We have added this information to the revised version of the manuscript (lines 70-72 of the methods section and lines 252-258 of the results section). 

The possible mechanisms that mediate the adverse effect of FGF23 in the patients in this study are not discussed. Although references from other preclinical studies are provided on the effect of FGF 23 on ventricular hypertrophy, renal fibrosis, proinflammatory effects and endothelial dysfunction.

This work can be an excellent opportunity to describe whether or not there is an association in transplant patients and shed more light on these mechanisms. Providing this information should not require much effort since, as the authors say in the section on strengths, they have detailed clinical and laboratory data.

Response: We thank the reviewer for the suggestion, however the aim of the study is to investigate whether the increased risk of EPO on death is to a large extent attributable to variation of FGF23. The request of the reviewer would require a different approach centered around FGF23. We already previously performed such a study, and refer the reviewer to this article from our group (Baia LC et al. Clin J Am Soc Nephrol 2013) regarding the association between FGF23 and death in this same cohort. Furthermore, the increased risk of death due to FGF23 has been shown in multiple studies in different patient groups, e.g. in RTR (Wolf M et al. J Am Soc Nephrol 2011; 22(5):956-66, Baia LC et al. Clin J Am Soc Nephrol 2013; 8(11):1968-78), in CKD patients (Isakova T et al. JAMA 2011; 305(23):2432-9), Scialla JJ et al. J Am Soc Nephrol 2013; 24(1):125-35), and healthy elderly (Ix JH et al. J Am Coll Cardiol 2012; 60(3):200-7). The exact mechanisms are not known how FGF23 leads to the increased risk of these adverse effects, but we have mentioned several possible effects that have been unraveled previously. The off-target effects range from causing renal fibrosis by binding to FGFR4 in renal fibroblasts, causing left ventricular hypertrophy by binding to FGFR4 in cardiac myocytes, exerting pro-inflammatory effects by increasing IL-6 en CRP expression by hepatocytes, and causing an impaired immune response by binding to FGFR1 in macrophages and FGF2 in neutrophils. Most likely, the increased risk of death attributable to FGF23 is caused by a combination of the listed effects. We have added this information to the revised version of the manuscript (lines 335-336 of the discussion section).

Reviewer 4 Report

Thank you for the opportunity to revise this interesting work. Some minor issues:

Why was ICD-9 used instead of the more recent ICD-10?

What were vitamin D levels?

I would discuss the role of phosphate, PTH and vitamin D and their reported association with cardiovascular risk in the light of these findings. 

Other markers of inflammation (i.e. cytokines, cell subtypes) have not been presented; it is possible that they might contribute explaining the results, so I would state this limitation.

Author Response

Thank you for the opportunity to revise this interesting work.

Response: We thank the reviewer for the kind words

Some minor issues:

Why was ICD-9 used instead of the more recent ICD-10?

Response: ICD-9 was default during the time moment that this observational data is collected from 2001 to 2003. Hence, we used the ICD-9 codes which have been reported in that time frame to diagnose the cause of death. As requested by reviewer #3, we have added the link to the ICD-9 codes in the revised version of the manuscript (line 69 of the methods section of the revised version of the manuscript).

What were vitamin D levels?

Response: We do not have vitamin D levels available in the whole cohort, only in a subset of the cohort, namely in 415 RTRs. In this subcohort, the mean 25-hydroxyvitamin D [25(OH)D] and 1,25-dihydroxyvitamin D [1,25(OH)2D] levels were 53±23 nmol/L and 109±46 pmol/L. Across quartiles of EPO, levels of both 25(OH)D and 1,25(OH)2D were not significantly different. To accommodate the comment of the reviewer, we have added this information to the revised Table 1 of the revised version of the manuscript, with mentioning in the foot legend that vitamin D levels were only available in a subset of the total cohort.

I would discuss the role of phosphate, PTH and vitamin D and their reported association with cardiovascular risk in the light of these findings. 

Response: We thank the reviewer for the comment. To accommodate the comment of the reviewer, we will present the results of these analyses here. Phosphate levels were not associated with all-cause death (HR, 1.28; 95%CI 0.49-3.34; P=0.62) and cardiovascular death (HR 3.47; 95%CI 0.92-13.2; P=0.06) after multivariable adjustment for general potential confounders (age, sex, body surface area, and eGFR). Similarly, PTH was not associated with all-cause death (HR, 1.09 0.78-1.52; P=0.63) and cardiovascular death (HR, 1.03; 95%CI 0.65-1.63) following adjustment for potential confounders. The main results of vitamin D levels have been described previously by our group (Keyzer CA et al. J Clin Endocrin Metab 2015). In our cohort within 415 RTRs having both EPO and vitamin D levels available, results are unaltered, i.e. low 25(OH)D is independently associated with an increased risk of all-cause and cardiovascular death. However, as we have stated in the previous comment by this reviewer, vitamin D levels were not different across quartiles of EPO (Table 1).

Other markers of inflammation (i.e. cytokines, cell subtypes) have not been presented; it is possible that they might contribute explaining the results, so I would state this limitation.

Response: We thank the reviewer for bringing up this point and indeed that is a limitation as we only adjusted for hs-CRP. As requested by the reviewer, we have added to the limitations section of the revised manuscript that we did not have other markers of inflammation (i.e. cytokines, cell subtypes) (line 355-357 of the discussion section).

Round 2

Reviewer 2 Report

In the revised version of the manuscript authors better described their results increasing clinical informations, improving statistical analysis and better describing the limitation of the study. I have no other suggestions.

Author Response

In the revised version of the manuscript authors better described their results increasing clinical informations, improving statistical analysis and better describing the limitation of the study. I have no other suggestions.

Response: We thank the reviewer for the comment and are glad to see that the reviewer states that in the revised version of the manuscript the results have increasing clinical information, that the statistical analysis has been improved and that we have better described the limitations of the study.

Reviewer 3 Report

The authors have answered most of the questions raised in the previous review and have submitted a manuscript more easily interpretable by readers.
Despite this, Table 4 continues to be somewhat confusing. A small change should be incorporated to improve its understandability:

Throughout the manuscript, the Hazard ratio has been used as a method to quantify the strength of the association of a factor with the outcome. However, in Table 4 the coefficient is used. Table 4 should also incorporate HR, either by replacing the coefficient with HR or by adding a column containing the values of HR and 95% CI. It is not a complex process: it is about calculating the exponentials of the coefficients.

Author Response

The authors have answered most of the questions raised in the previous review and have submitted a manuscript more easily interpretable by readers.
Despite this, Table 4 continues to be somewhat confusing. A small change should be incorporated to improve its understandability:
Throughout the manuscript, the Hazard ratio has been used as a method to quantify the strength of the association of a factor with the outcome. However, in Table 4 the coefficient is used. Table 4 should also incorporate HR, either by replacing the coefficient with HR or by adding a column containing the values of HR and 95% CI. It is not a complex process: it is about calculating the exponentials of the coefficients.

Response: We thank the reviewer for the comment. As requested by the reviewer, we have added to the mediation Table an explanation how the reader is able to interpret quickly the identified coefficients back to odds ratios since as we had described in the manuscript we performed mediation analysis according to Preacher and Hayes which is based on logistic regression. To calculate the coefficient back to odds ratios, we have added in the Table 4 the unstandardized coefficient for total effect as determined in mediation analysis according to Preacher and Hayes and explained in the foot legend of Table 4 how the coefficient can be calculated back to odds ratios to increase the interpretation possibilities of the readers. The unstandardized coefficient of the total effect can be calculated to the related odds ratio as eβ. This largely corresponds to the hazard ratio of the Cox regression analysis, although there is always difference due to the fact that the former does not take time-to-event into account whereas the latter does. For example, the unstandardized coefficient of the total effect of EPO on all-cause death, while adjusting for FGF23 is 0.120, which can be calculated to an OR by taking the exponent of this regression coefficient, i.e. e0.120=1.12, which would be the equivalent of HR 1.28 (Table 2; model 5) while not taking into account the time factor. We added this information to the revised version of the manuscript (lines 264-269 of the results section). With respect to the percentage mediation that occurs, it should be realized that 100 multiplied by β1/β2 is not the same as 100 multiplied by eβ1/eβ2, therefore to avoid confusion for the readers we maintained currently the percentage mediation as identified with the mediation analysis according to Preacher and Hayes. In addition, to further accommodate the comment of the reviewer, we have added also as secondary analyses to the revised version of the manuscript the percentage change of the hazard ratio prior to and following adjustment for FGF23. The percentage change in hazard ratio has been calculated as follows: (HR before adjustment – HR after adjustment)/(HR before adjustment – 1) x 100% (Oterdoom LH et al. Transplantation 2009). If we calculate the percentage change of HR following adjustment for FGF23, we identified that adjustment for FGF23 was responsible for a 58% reduction in HR for the association between erythropoietin and all-cause death and 48% reduction in HR for the association between erythropoietin and cardiovascular death. We have added this information to the revised version of the methods section (lines 135-137) and to the revised version of the results section (lines 249-251) of the manuscript.